# Hypothalamus and Post-Traumatic Stress Disorder: A Review

**DOI:** 10.3390/brainsci13071010

**Published:** 2023-06-29

**Authors:** Payman Raise-Abdullahi, Morvarid Meamar, Abbas Ali Vafaei, Maryam Alizadeh, Masoomeh Dadkhah, Sakineh Shafia, Mohadeseh Ghalandari-Shamami, Ramtin Naderian, Seyed Afshin Samaei, Ali Rashidy-Pour

**Affiliations:** 1Research Center of Physiology, Semnan University of Medical Sciences, Semnan, Iran; payman.raise.abdullahi@semums.ac.ir (P.R.-A.);; 2Department of Physiology, School of Medicine, Semnan University of Medical Sciences, Semnan, Iran; 3Department of Basic Medical Sciences, Faculty of Medicine, Qom Medical Sciences, Islamic Azad University, Qom, Iran; 4Pharmaceutical Sciences Research Center, Ardabil University of Medical Sciences, Ardabil, Iran; 5Immunogenetics Research Center, Department of Physiology, Mazandaran University of Medical Sciences, Sari, Iran; 6Department of Physiology, Faculty of Medicine, Tehran Medical Sciences, Islamic Azad University, Tehran, Iran; 7Student Research Committee, Semnan University of Medical Sciences, Semnan, Iran; 8Department of Neurology, School of Medicine, Semnan University of Medical Sciences, Semnan, Iran

**Keywords:** hypothalamus, PTSD, HPA axis, HPT axis, HPG axis, growth hormone, prolactin, oxytocin, vasopressin

## Abstract

Humans have lived in a dynamic environment fraught with potential dangers for thousands of years. While fear and stress were crucial for the survival of our ancestors, today, they are mostly considered harmful factors, threatening both our physical and mental health. Trauma is a highly stressful, often life-threatening event or a series of events, such as sexual assault, war, natural disasters, burns, and car accidents. Trauma can cause pathological metaplasticity, leading to long-lasting behavioral changes and impairing an individual’s ability to cope with future challenges. If an individual is vulnerable, a tremendously traumatic event may result in post-traumatic stress disorder (PTSD). The hypothalamus is critical in initiating hormonal responses to stressful stimuli via the hypothalamic–pituitary–adrenal (HPA) axis. Linked to the prefrontal cortex and limbic structures, especially the amygdala and hippocampus, the hypothalamus acts as a central hub, integrating physiological aspects of the stress response. Consequently, the hypothalamic functions have been attributed to the pathophysiology of PTSD. However, apart from the well-known role of the HPA axis, the hypothalamus may also play different roles in the development of PTSD through other pathways, including the hypothalamic–pituitary–thyroid (HPT) and hypothalamic–pituitary–gonadal (HPG) axes, as well as by secreting growth hormone, prolactin, dopamine, and oxytocin. This review aims to summarize the current evidence regarding the neuroendocrine functions of the hypothalamus, which are correlated with the development of PTSD. A better understanding of the role of the hypothalamus in PTSD could help develop better treatments for this debilitating condition.

## 1. Introduction

Trauma is an experience that can cause pathological metaplasticity, leading to long-lasting behavioral changes and impairing an individual’s ability to cope with future challenges. In essence, the experience of trauma can result in altered neural responses that compromise an individual’s capacity to adapt effectively to new situations [1]. Post-traumatic stress disorder (PTSD) is a common and debilitating psychiatric disorder usually characterized by a response to a traumatic event, such as intense fear, panic, or helplessness [2]. Exposure can be actual or threatened death, serious injury, or sexual violation through direct personal experience or witnessing an “extremely threatening or horrific event or series of events” [3]. Critical diagnostic features of PTSD include intrusive and recurring memories, avoidance, general emotional numbness, alterations in arousal and reactivity, functional impairment in social or occupational contexts or distress, and others [2]. Most intrusive memories include recurring thoughts, images, perceptions, hallucinations, and dreams related to the traumatic event. Emotional numbing also manifests as attempts to avoid recalling stimuli, inability to recall event details, decreased interest or participation in activities, and social withdrawal from others. Hyperarousal symptoms include insomnia, irritability, hypervigilance, exaggerated startle response, and anger. In addition, patients with PTSD often have cognitive function deficits, including attention, learning, and memory impairment [4,5].

PTSD may occur in individuals of any age, ethnicity, nationality, or culture. In the United States, approximately 3.5% of adults are affected by PTSD each year, and among adolescents aged 13–18, the lifetime prevalence is 8%. Its prevalence is estimated to be 5–8% in men and 10–14% in women [6,7]; so, women are twice as likely as men to develop PTSD. In addition, PTSD is more prevalent in high-risk professional groups, such as military service workers [8], and some ethnic groups are disproportionately affected by this condition and experience higher rates of PTSD compared to non-Latino whites [9]. 

PTSD is a complex and highly variable disorder that affects a significant portion of the population. Due to the destructive nature of PTSD and the insufficient treatment options available for patients diagnosed with this condition, research has focused on early pharmacological interventions following traumatic events. Previous studies indicate that administering certain medications, such as morphine [10,11], propranolol [12], hydrocortisone and dexamethasone [13], and oxytocin [14], after a traumatic event may shield individuals from developing PTSD or lessen the severity of their symptoms. In addition, various intervention approaches, such as cognitive behavioral therapy (CBT) and prolonged exposure (PE) therapy, have been developed to address the diverse needs of individuals with PTSD. PE is a highly effective form of treatment for those who have PTSD as it consistently improves conditions sustained over time. PTSD affects various subpopulations differently, and the effectiveness of intervention approaches can vary accordingly. For example, combat and non-combat veterans may experience different outcomes following PTSD treatment. Research has shown that CBT effectively reduces PTSD symptoms in veterans, with prolonged exposure therapy particularly effective in 60% of veterans with PTSD [15]. However, a significant minority did not show improvements in PTSD severity 12 months after treatment, highlighting the need for further research to understand the factors influencing treatment outcomes in this subpopulation. Although the first-line trauma-focused interventions, CPT and PE, have demonstrated clinically significant progress in treating PTSD among military and veteran populations, nonresponse rates continue to be high, and outcomes are only marginally better than active control conditions. As a result, there is a pressing need for improved existing PTSD treatments and innovative, evidence-based therapies, both trauma-focused and non-trauma-focused, to enhance treatment efficacy [16].

Adult sexual assault survivors also face unique challenges in recovering from PTSD. The risk factors for developing PTSD after experiencing rape differ from those associated with PTSD following a non-sexual assault [17]. Measuring recovery from PTSD resulting from sexual assault involves more than just the absence of symptoms or achieving particular outcomes [18]. Early intervention is critical for sexual assault victims, as the level of distress immediately following the assault strongly correlates with future pathologies and PTSD. However, there is no “cookie-cutter” treatment for every victim with PTSD, as the disorder can manifest in various ways. Recognizing the biological, psychological, and sociological implications is crucial while devising effective treatment and intervention strategies for PTSD resulting from sexual assault.

Furthermore, individuals with childhood trauma may also experience different outcomes following PTSD treatment. Childhood sexual trauma is associated with PTSD, depression, suicide, alcohol problems, and eating disorders. Some psychological interventions commonly used to treat PTSD in adults are effective in younger populations, but not all [19]. Trauma-focused cognitive behavioral therapy (TF-CBT) has been the most commonly researched intervention for children, adolescents, and young adults. Early interventions to prevent PTSD in youth after exposure to a potentially traumatic event have been shown to decrease the risk of PTSD through cognitive behavioral treatments [20].

The precise underlying reasons behind PTSD are not comprehensively comprehended, but changes in the brain subregions, including in the hypothalamus, may play a critical role [21]. The hypothalamus is a small but vital brain structure regulating various bodily functions, including the stress response. Following a stressful event, the hypothalamus activates the sympathetic nervous system and hypothalamic–pituitary–adrenal (HPA) axis, releasing hormones such as adrenaline and cortisol that help the body respond to the threat. Dysregulation of the hypothalamus may contribute to an excessive or prolonged stress response, which can lead to the development of PTSD [21]. Although the precise mechanisms by which the hypothalamus is involved in PTSD development are not yet fully understood, studies suggest that dysregulation of the hypothalamus and its interactions with other brain structures may contribute to the development of symptoms [21].

## 2. Hypothalamus Neuroanatomy and Main Functions

The hypothalamus is a crucial structure that coordinates behaviors essential for individual and collective survival. It integrates suitable autonomic and neuroendocrine responses to maintain homeostasis, serving as a highly integrated sensory and motor output region that significantly controls endocrine, autonomic, and somatic behaviors [21,22]. The hypothalamus receives internal stimuli via receptors for circulating hormones and neural pathways and plays a crucial role in limbic sensory integration via the fornix and mammillothalamic tract in the Papez circuit and the stria terminalis connection to the amygdala. The medial forebrain bundle receives cortical-level sensory perceptions, while the retinohypothalamic tract carries light signals to the suprachiasmatic nucleus, which regulates the diurnal pattern of hormone release. The hypothalamus integrates all these inputs, resulting in appropriate physiological and behavioral responses that ensure the maintenance of life over time [22].

The hypothalamus regulates endocrine and autonomic systems as well as somatic behaviors. The paraventricular and supraoptic nuclei produce oxytocin and antidiuretic hormone (ADH) peptides [23]. The preoptic, anterior, and posterior nuclei regulate body temperature, while the preoptic nucleus in males and females shows variations in estrogen receptor distribution, influencing sexual and maternal behavior [24]. The suprachiasmatic nucleus diurnally modulates hormone secretion and behavior based on the light input received through the eyes, leading to cortisol peaks around sunrise and growth hormone peaks near midnight [25]. The ventromedial nucleus regulates feeding behavior, whereas the dorsomedial nucleus controls rage. The lateral hypothalamus is responsible for sensing hunger and stimulating eating behaviors.

The arcuate nucleus releases hormones secreted by axon terminals into the hypothalamohypophysial venous portal system to regulate the release of anterior pituitary hormones. The corticotropin-releasing hormone (CRH) released by the arcuate nucleus stimulates the adrenocorticotropic hormone (ACTH) released by the anterior pituicytes into capillaries that eventually drain into the cerebral venous system. ACTH then travels through systemic circulation and prompts the adrenal cortex (zona reticularis) to produce cortisol, a stress-response hormone. Due to its diverse inputs, the hypothalamus enables the body to respond to physiological and psychological stressors. Cortisol production varies throughout the day, with the highest levels observed around sunrise and the lowest levels around sunset due to the interconnections between the arcuate and suprachiasmatic nuclei [26].

Glucocorticoids play a crucial role in regulating neuronal activities [27]. In addition to their feedback actions on hypothalamic-pituitary peptidergic cells, they also affect the function of several other neurotransmitters and neuromodulators. Research indicates notable differences in HPA axis activity in individuals with PTSD compared with those without PTSD or other psychiatric disorders [28]. For instance, studies demonstrate that individuals with PTSD experience enhanced suppression of cortisol levels after receiving synthetic glucocorticoid dexamethasone compared to non-psychiatric controls and trauma survivors exposed to similar conditions without PTSD [29]. People with PTSD also exhibit lower plasma and urinary cortisol levels and higher glucocorticoid receptor expression in lymphocytes [30]. These unique glucocorticoid-related changes suggest that individuals who develop PTSD have distinct pathophysiological alterations [31]. 

PTSD is associated with changes in several neuroendocrine systems beyond the HPA axis, as supported by numerous studies. Abnormalities in noradrenergic, dopaminergic, and serotonergic pathways have been documented in PTSD [32], as well as changes in other endocrine systems such as the hypothalamic–pituitary–thyroid (HPT) axis [33] and the hypothalamic–pituitary–gonadal (HPG) axis [34]. These alterations are thought to contribute to the diverse symptomatology of PTSD, including emotional dysregulation, hyperarousal, and sleep disturbances. Moreover, these findings suggest that PTSD is a complex disorder that involves dysregulation of multiple neuroendocrine systems rather than being solely related to HPA axis dysfunction. 

The thyrotropin-releasing hormone (TRH) is vital in regulating the HPT axis, which controls metabolism and other homeostatic functions by regulating the blood levels of thyroid hormones. Trauma has been known to trigger thyroid abnormalities, and studies have suggested a possible relationship between the HPT axis and PTSD. Research has shown that individuals with PTSD, including Vietnam and WWII veterans, had elevated baseline levels of both tri-iodothyronine (T3) and thyroxine (T4), with T3 levels being disproportionately elevated relative to T4, indicating an increase in the peripheral deiodination process [35,36]. However, there is controversy, and more studies are needed to fully understand the relationship between the HPT axis and PTSD, including the direction of causality and the potential mechanisms underlying this relationship.

Glucocorticoids can directly impact neurotransmitters, subsequently modulating prolactin release [37]. Animal studies suggest that corticosteroid administration can reduce the synthesis and release of prolactin from the pituitary gland, and this effect can be reversed through adrenalectomy [38]. In humans, research investigating peripheral dopamine levels in PTSD has shown increased urinary and plasma dopamine concentrations [39,40]. Moreover, combat veterans with PTSD exhibit reduced platelet serotonin uptake [41]. In addition to the HPA axis, other neuroendocrine systems related to the hypothalamus, such as the HPT axis, prolactin, and sex hormones, also play a role in PTSD. Therefore, it is important to consider the involvement of these systems in the pathophysiology of PTSD to develop more comprehensive treatments for the disorder.

## 3. Hypothalamic–Anterior Pituitary–Adrenal Gland Axis and PTSD

The prefrontal cortex (PFC) and limbic structures, especially the amygdala and hippocampus, interact highly with the hypothalamus. They examine the disruption in homeostasis the person is experiencing, compare it to past experiences, and categorize it as a stressful or normal situation [21]. Identifying a stressful stimulus triggers the stress response. Several stress mediators, including hormones, neuropeptides, and neurotransmitters, are involved in the stress response. The hypothalamus, linked to the above-mentioned regions, acts as a central hub, integrating physiological aspects of the stress response [21]. Following a stressful situation, the hypothalamus activates two pathways: the autonomic nervous system (ANS) and the HPA axis. The first one is faster, releasing adrenaline and noradrenaline from the adrenal medulla. Whereas the second pathway, which acts slower, leads to the release of glucocorticoids from the adrenal cortex (Figure 1). HPA axis dysregulation is correlated with some mental disorders, such as PTSD, depression, and schizophrenia [42,43]. Several other stress mediators, including hormones, neuropeptides, and neurotransmitters, are involved in the stress response in cooperation with previous pathways [21]. Moreover, some other monoamines, such as dopamine and serotonin, released from different brain areas, are involved in behavioral aspects of the fast stress response [21]. 

Neuropeptides, including orexin, ghrelin, dynorphin, urocortin, oxytocin, neuropeptide Y, galanin, and substance P, affect stress response [21,44,45]. However, the principal neuropeptides modulating stress response are CRH, also known as corticotropin-releasing factor (CRF), and arginine vasopressin (AVP), also known as vasopressin, or antidiuretic hormone (ADH), first detected in the hypothalamus [46,47]. Neurons located in the median eminence of the hypothalamus release CRH in stress response [46]. However, it has been found in other regions, such as the amygdala, hippocampus, and the locus coeruleus [48,49,50]. Soon after release, CRH binds to its G protein-coupled receptors, CRHR1 and CRHR2, affecting neuronal firing patterns, gene expression, and behaviors [21]. Other neuropeptides, such as urocortin, can bind to CRHR and thus may play a role in stress adaptation [51]. 

CRH, the initiating part of the HPA axis, is released mainly from the hypothalamus. However, this neuropeptide may be found in other brain areas, exerting various effects. For instance, CRH, secreted from the central amygdala, binds to the basolateral amygdala CRHR1 [52] and CRHR2 [53] and may modulate stress-emotional memories and anxiety [54,55]. Moreover, it is involved in terminating stress-related anxiety [54]. In the locus coeruleus, CRH binds to CRHR1 [56,57], interacting with noradrenaline pathways [50]. CRH released by hippocampal interneurons binds to pyramidal cells CRHR1 [44,58], affecting stress-related learning and memory [59]. CRH, released by the bed nucleus of the stria terminalis (BnST), modulates stress-related anxiety [48,60] via CRHR1 located in the nucleus accumbens [61] and CRHR2 [53,57] (Figure 2). 

Vasopressin is secreted by the hypothalamic dendrites, BnST, and amygdala [47,48,62]. CRH and AVP in the hypothalamus jointly affect the anterior pituitary to release ACTH [63]. ACTH drives the adrenal cortex to produce glucocorticoid hormones (cortisol in humans, most fish, and mammals such as dogs and hamsters; corticosterone in rats, mice, birds, and most reptiles) [64]. Cortisol and corticosterone are critical components of the fear and stress physiological response [6], acting via two intracellular receptors: mineralocorticoid receptors (MRs) and glucocorticoid receptors (GRs). MRs are primarily saturated in basal conditions. GRs are mainly occupied when glucocorticoid circulating levels are high (during the stress and peak of circadian rhythms) [65] and are involved in fear responses and fear memory extinction [66,67,68,69].

Adrenalin and noradrenaline promote the fight and flight response, whereas glucocorticoids induce long-term metabolic processes into short-term survival functions. CRH and ACTH, and consequently cortisol, fluctuate in the circadian cycle. Cortisol culminates following an hour of waking, i.e., cortisol awakening response (CAR). Then, it declines during the day (diurnal cortisol slope) and reaches its lowest in sleep [70]. High plasma cortisol levels inhibit CRH and ACTH by a negative feedback loop. The short-term catabolic functions should not last long and must be terminated by the negative cortisol feedback on the CRH and AVP [21]. Any alterations in diurnal cortisol levels may dysregulate the HPA axis, impairing several aspects of homeostasis, such as glucose metabolism, digestion, and inflammation, as well as learning and memory [71,72].

The HPA axis correlates directly with PTSD, but how? The HPA axis and cortisol levels in PTSD came into focus after Mason and Giller first reported lower urinary cortisol levels in patients with PTSD than in those with major depression (MDD) and paranoid schizophrenia [30]. Several studies reported low cortisol levels in the urine of PTSD patients [73,74]. A study on bereaved children following the September 11 attack showed that baseline cortisol was low and cortisol suppression was greater in bereaved children with PTSD compared to the non-PTSD control subjects [75]. Goenjian et al. found low cortisol levels in the saliva samples and greater cortisol suppression following dexamethasone in earthquake victims [76]. Roth et al. have shown that saliva cortisol levels were low in mass-evacuated adults from Kosovo [77]. However, PTSD and cortisol level correlation seems inconsistent. Bonne et al. reported similar mean cortisol levels in PTSD and non-PTSD subjects one week after traumatic events [78]. Young et al. did not find saliva cortisol changes associated with PTSD [79]. Baker et al. reported higher CSF cortisol levels but not plasma ACTH or peripheral cortisol in PTSD patients. 

Sillivan et al. used a rodent model to investigate PTSD and found that inbred stressed mice had elevated plasma corticosterone levels [80]. In a similar study, Torrisi et al. utilized a translational animal model of PTSD to investigate plasma corticosterone levels. They restrained mice for 24 h and found that the amount of plasma corticosterone was significantly higher in the long-term post-trauma animals compared to the control group [81]. Danan et al., using predator scent stress as an animal model of PTSD, discovered that an individual’s pre-existing susceptibility to the disorder could be identified through a blunted basal corticosterone pulse amplitude. They measured hormone secretion patterns by collecting blood samples at 20-min intervals and evaluated behavioral responses. The results showed that the animals with a PTSD-like phenotype had significantly reduced ultradian oscillations of corticosterone levels eight days later, along with a blunted corticosterone response to stressors and extreme behavior disruption. These findings suggest that blunted basal corticosterone pulse amplitude may serve as a biomarker for susceptibility to PTSD [82].

The inconsistencies in the literature regarding HPA axis dysregulation in PTSD may be due to the physiological complexities of the HPA axis and the different methods used to assess it. Other possible factors include the method used to measure cortisol, the types of samples collected (urine, blood, saliva, hair, or cerebrospinal fluid), the timing of sample collection about the traumatic event, genetic and epigenetic factors, the severity of illness, sex, alcohol or tobacco use, age, medical conditions, and comorbid psychiatric disorders [6,72,83,84,85,86,87]. 

In conclusion, while the HPA axis appears to be dysregulated in individuals with PTSD, it cannot be definitively stated that HPA axis dysregulation is the cause of PTSD or is caused by PTSD (Figure 1). A malfunctioning HPA axis before and after traumatic events seems to be a risk factor for developing PTSD. Although the underlying mechanism of PTSD is primarily associated with excessive feedback of the HPA axis and biological imbalances in fear-related brain circuits [88], further research is needed to fully understand the complex interplay between these factors and how they contribute to developing and maintaining PTSD.

## 4. Hypothalamic–Anterior Pituitary–Thyroid Gland Axis and PTSD

PTSD is associated with multiple hormonal and neurotransmitter alterations, including cortisol [89,90,91], serum lipids [92], noradrenaline, serotonin [39], and thyroid hormones [93,94]. While the role of the HPT axis in PTSD is poorly understood, there have been explorations of the connections between traumatic experiences and thyroid function. Classically, the HPT axis plays a vital role in growth, differentiation, metabolism [95,96], sexual behavior [97], and evolution [98]. Alterations in thyroid hormone levels have been linked to psychiatric diseases [95,96]. The onset of hyperthyroidism after a terrifying experience was described by Parry in 1825, and Bram reported a clear connection with traumatic stress in the early 20th century in more than 85% of over 3000 cases of thyrotoxicosis [99]. The HPT axis is controlled by TRH, which is synthesized in hypothalamic paraventricular nucleus neurons and released near portal vessels in the median eminence before being transported to the anterior pituitary. There, TRH activates TRH receptor type 1 (TRH-R1) in thyrotrophs, increasing the synthesis and release of thyrotropin (thyroid-stimulating hormone, TSH). TSH controls the synthesis of thyroid hormones T4 and T3, and T4 is converted to T3 by tissue deiodinases I or II [100,101,102,103].

Energy-related cues and stressors regulate the activity of the HPT axis, and there is a complex interaction between environmental and physiological factors in its regulation, including neonatal stress, nutrition, and sex. It has been shown that neuroendocrine responses to stress and nutrition are sexually dimorphic [104,105,106,107,108], and maternal separation causes long-term changes in the offspring in a sex-specific manner on some elements of the HPT axis in adult rats [109,110]. Studies investigating the relationship between trauma/PTSD and thyroid function can be divided into two main groups: male soldiers and refugees [33,36,95,111,112,113,114] and studies of females exposed to childhood physical or sexual abuse [108,115,116]. Both hyper and hypothyroidism are associated with PTSD in these studies. 

Elevations in serum-free and total triiodothyronine (FT3 and T3) levels have been reported in World War II veterans, Vietnam combat veterans with combat-related PTSD, and PTSD women with experiences of sexual abuse compared with healthy subjects. Increased serum T4 levels have also been linked to stress arousal, indicating the involvement of the HPT axis in stress regulation and stress-related disorders [36,93,94,113,115]. However, other studies have reported the opposite results. For example, decreased thyroid hormone levels have been reported in East German refugees with psychiatric diseases, including PTSD [111]. Chronic and some forms of acute psychological stress can inhibit HPT axis activity, even without a history of early-life stress [100,103,104,117,118,119,120,121]. A history of physical/emotional abuse or neglect has been associated with reduced plasma levels of T3 in adolescents [105] and low plasma levels of TSH in adult women [106]. A study of American veterans in Iraq and Afghanistan found that individuals with PTSD had an increased risk of autoimmune thyroiditis compared with those without psychiatric disorders [122]. Other psychiatric disorders were associated with a 40% increased risk of thyroiditis, while PTSD was associated with a 92% risk of thyroiditis. However, it is unclear whether the thyroiditis was hyperthyroidism or hypothyroidism. Trauma/PTSD has been associated with changes in thyroid hormone levels [33,36,93,108,111,112,113,115,116,123,124]. The mechanisms underlying the association between PTSD and hypothyroidism are poorly understood; however, trauma may reduce thyroid hormone levels over a long period [120,125,126,127].

The conflicting results of studies examining the relationship between trauma/PTSD and thyroid function can be attributed to several factors. Firstly, the study populations differed in gender, age, and diagnosis. Secondly, the trauma exposure’s type, timing, and duration varied across studies [108,128]. The thyroid system can respond to stress quickly [35,116], with transient activation of the HPT axis observed after exposure to acute stress, leading to increases in circulating TSH due to the direct stimulatory effect of glucocorticoids on the pituitary thyrotrophs [129]. However, prolonged stress and prolonged exogenous administration or endogenous overproduction of glucocorticoids are accompanied by decreased HPT activity in humans and experimental animals [129,130,131]. Both metabolic [132,133] and psychological [117] stressors can affect the activity of both the HPA and HPT axis, resulting in decreased thyroid hormones, activation of the HPA, and release of glucocorticoids. Glucocorticoids can inhibit the HPT axis at the level of the hypothalamus and pituitary and can also inhibit the peripheral conversion of T4 to T3 [130], resulting in stress-induced decreases in serum T3 levels [134]. However, some in vitro studies have shown that glucocorticoids can stimulate TSH production [135,136].

Enhanced somatostatin secretion due to increased CRH release during stress may decrease TSH secretion. However, somatostatin is a weaker inhibitor of TSH secretion than growth hormone [137]. Glucocorticoids do not affect the stimulatory effect of exogenous TSH on thyroid hormone production and turnover or thyroxin-binding globulin levels [131,138]. Cytokine factors are also involved in HPT activity inhibition by stress [139]. They decrease THS bioactivity by preventing glycosylation and inhibiting deiodinase-2 in peripheral tissues [129]. Different thyroid function profiles may be related to different coping responses to traumatic stress [35,140]. More robust acute responses to stress can cause a decrease in thyroid hormones, as seen in the HPA axis, which may explain the hyperarousal symptoms of PTSD [35,140]. However, another study showed a strong correlation between thyroid hormone levels and hyperarousal symptoms of PTSD [33]. Noradrenaline levels have been found to correlate directly with the number of traumatic events in veterans with PTSD, with soldiers who experienced more traumatic events having higher levels of T3 [39,113]. Immobilization stress has been shown to both increase and decrease thyroid hormone levels [117,141,142,143]. Noise stress increases TSH levels, while lipopolysaccharide (LPS) injection [144] and inescapable tail shock [145] reduced levels of thyroid hormones. The products of the HPT axis also affect the HPA axis, with the elimination of thyroid hormones leading to a decrease in CRH mRNA within the hypothalamus [146]. The inhibitory effect of thyroid hormones on HPA axis activity has also been reported [147]. A corticotropin release-inhibiting factor is proposed to be located within the prepro-TRH peptide [148]. In addition to HPA axis products, hypothalamic TRH levels are regulated by other peptides, including melanocyte-stimulating hormone (MSH), neuropeptide Y (NPY), and agouti-related protein (AGRP) located within the arcuate nucleus and PVN of the hypothalamus [133,149]. 

In conclusion, the altered functioning of the HPT axis may contribute to developing and maintaining PTSD symptoms. Thyroid hormones play a role in regulating mood and cognition, and changes in thyroid hormone levels may contribute to developing PTSD. The relationship between the HPT axis and PTSD is complex and not well understood. Some studies have found increased, decreased, or no differences in thyroid hormone levels in individuals with PTSD, while others have found that alterations in the HPT axis may be more pronounced in individuals with more severe PTSD symptoms. Both hyperthyroidism and hypothyroidism, as well as thyroiditis, have been associated with PTSD. However, increased T3 levels may be involved in hyperarousal, which is a common symptom of PTSD. Further research is needed to fully understand the complex interplay between the HPT axis and PTSD, including the potential involvement of other factors such as genetics, environmental influences, and sex.

## 5. Hypothalamic–Anterior Pituitary–Gonads Glands Axis and PTSD

In addition to the HPA and HPT axes, the HPG axis and gonadal steroid hormones also may modulate stress-related responses in individuals with PTSD [150,151] (Figure 3). Stress has been identified as a significant disrupting factor in the reproductive system and sexual behavior [152,153]. This connection is regulated by the HPG axis since there is close communication between the HPG and HPA axes [151]. The HPG axis is closely linked to the HPA axis, and communication between the two systems is essential for maintaining the balance between reproduction and survival [154]. Corticosterone, the final product of the HPA axis, can inhibit the HPG axis, leading to dysfunction of the sexual and reproductive system under stress [155]. Conversely, testosterone, a product of the HPG axis, can inhibit the HPA axis. The HPG axis modulates the HPA axis through the gonadal steroid hormones estradiol (E2), progesterone (P4), and testosterone [156,157]. For example, testosterone inhibits CRH, reducing cortisol secretion in humans [158], and suppresses ACTH and corticosterone production in rodents [159].

Abnormal levels of androgens have been observed in various psychiatric disorders [160,161], with circulating testosterone levels being reduced by both physical and psychological stress [162,163,164]. Numerous studies have reported lower testosterone levels in other psychiatric disorders, such as major depressive disorder (MDD), with a negative correlation between testosterone levels and the severity of depression [165,166,167,168]. However, some studies have shown that testosterone levels increase during potentially stressful events [169,170,171]. Compared to the HPA axis, the role of the HPG axis in PTSD has not been extensively studied. However, the pattern of testosterone levels in individuals with PTSD appears somewhat inconsistent. While some studies have found lower testosterone levels in individuals with PTSD [172], others have reported no differences or higher values [34]. For example, although Bauer et al. found no differences in testosterone levels of refugees compared to healthy subjects [111], Mason et al. demonstrated that testosterone levels were significantly higher in the PTSD group compared to the MDD patients. They also suggested that chronic basal testosterone levels in PTSD patients may be elevated compared to normal subjects [34]. Moreover, Karlović et al. researched soldiers with PTSD and found no significant differences in serum testosterone levels compared to control subjects. However, when they specifically examined PTSD patients without taking into account comorbidities such as MDD and alcohol dependence, the results showed elevated testosterone levels in the PTSD group [173]. 

HPG axis hormones are also essential in the formation of the structure and function of the brain. These hormones suppress atrophy and neurodegeneration induced by stress hormones in different brain parts [174,175,176] and induce neuronal plasticity and synaptic remodeling [177]. Moreover, testosterone improves mood and behavior and reforms cognitive skills [178]. Stress exposure downregulates testosterone and estradiol receptors in the hippocampus. Exogenous testosterone administration after exposure to trauma can decrease stress responses [179]. Testosterone is converted to dihydrotestosterone (DHT) and estradiol, acting on androgen and estradiol receptors. Testosterone induces its effect via genomic and non-genomic pathways [178] and exerts anxiolytic and anti-depressant effects by these receptors [180,181,182]. The hippocampus seems to be the main region that testosterone affects [179]. It is suggested that the influence of testosterone in modulating mood and anxiety is applied by the interaction of CRH receptors and androgen receptors [157].

Stress can disrupt the synthesis and secretion of gonadotropins and ovarian cyclicity [35,36]. Exogenous corticosterone has been shown to suppress the pulsatile pattern of luteinizing hormone (LH) release due to the inhibition of upstream gonadotropin-releasing hormone (GnRH) pulsatile secretion [183,184,185]. Kisspeptin neurons may regulate the alteration in HPA axis hormones in the arcuate nucleus (ARC) and anteroventral periventricular nucleus, and neighboring periventricular nucleus (AVPV/PeN) regions, as well as the dorsal-medial nucleus of the hypothalamus (DMN) RFamide-related peptide-3 (RFRP-3) neurons, which are mammalian orthologues of gonadotropin inhibitory hormone (GnIH) neurons [186,187]. However, the cellular mechanisms underlying this function are still unclear. Studies have demonstrated that increasing the activity of RFRP neurons in female mice following restraint stress can inhibit GnRH and LH output [186,187]. 

Gender is a significant risk factor for developing PTSD [188,189]. Women are more susceptible to PTSD following a traumatic event, although this can also be influenced by life history, personality, and the type of trauma experienced [190]. Compared to non-pregnant women, pregnant women exhibit greater clinical and psychophysiological hyperarousal [191]. In addition, some evidence correlates estrogen with intrusive memories, a characteristic of PTSD [192], suggesting that women who experience trauma with higher estrogen levels may face an increased risk of developing intrusive memories. Some observations indicate that basal corticosterone levels are higher in females than males, and the restraint stress model leads to more glucocorticoid release in female animals than in males [157]. These results suggest that estradiol increases HPA axis reactivity, although the findings are inconsistent, likely due to the diverse functions of estradiol receptors, including E2α and E2β [154]. There are some contradictory findings. The failure to inhibit fear responses in a safe situation may serve as the PTSD biomarker, and poor retention of extinction learning is considered one of the underlying causes of PTSD. The retention of previously conditioned fear responses in women may be linked to their estrogen and progesterone levels. Low estrogen levels in women may impair fear extinction and, as Glover et al. have suggested, be a risk factor for developing PTSD in women with trauma histories [193]. Moreover, Pineles et al. demonstrated that during the mid-luteal phase of the menstrual cycle, women with PTSD show disturbed retention of extinction learning compared to control subjects [194]. 

In contrast to estradiol, testosterone reduces HPA axis reactivity by binding to androgen receptors in hypothalamic reproductive areas such as the ARC, ventromedial nucleus of the hypothalamus (VMN), and preoptic nuclei [157]. Gonadectomized male rats show elevated corticosterone release after exposure to stressors compared to gonad-intact males, which diminishes after administering testosterone or DHT [159]. Progesterone and its metabolite, allopregnanolone (ALLO), may modulate stress responses [156]. An increase in progesterone and ALLO concentrations can depress HPA axis signaling. This inhibition mechanism is mediated by the binding of ALLO to gamma-aminobutyric acid (GABA) receptors. ALLO can suppress CRH gene transcription, inhibiting HPA axis activity [156]. Moreover, it is suggested that PTSD may impair the synthesis of progesterone metabolites, including ALLO and pregnanolone, which are involved in the facilitation of GABA_A_ receptors [195].

In summary, the HPG axis and gonadal steroid hormones may modulate stress-related responses in individuals with PTSD. Stress can disrupt the synthesis and secretion of gonadotropins and ovarian cyclicity, and the HPG axis also plays a role in the formation of the structure and function of the brain. Abnormal levels of gonadal steroid hormones have been observed in various psychiatric disorders, including PTSD and MDD. Both physical and psychological stress may reduce testosterone levels. It is suggested that testosterone exerts anxiolytic and antidepressant effects via interaction with CRH receptors and androgen receptors. Gender is also a significant risk factor for developing PTSD, with women being more susceptible to PTSD following a traumatic event. Reportedly, basal corticosterone levels are higher in females than males, and the restraint stress model leads to more glucocorticoid release in female animals than in males, suggesting estradiol may increase HPA axis reactivity. Moreover, some evidence correlates estrogen with intrusive memories, a characteristic of PTSD. However, low estrogen levels in women were correlated to fear extinction impairment. Gonadal hormones affect extinction learning retention differently in women with and without PTSD. Women with PTSD display poorer extinction retention during the mid-luteal phase of the menstrual cycle compared to trauma-exposed women without PTSD, who retain extinction learning better during this phase than in the early follicular phase. Progesterone and its metabolite, ALLO, modulate stress responses and can depress HPA axis signaling. PTSD may impair the synthesis of progesterone metabolites, including ALLO and pregnanolone. Again, the correlation between testosterone levels and PTSD seems inconsistent. While some studies have found lower testosterone levels in individuals with PTSD, others have reported no differences or even higher values. Overall, the complex interplay between the HPA and HPG axes and gonadal steroid hormones highlights the importance of further research to better understand the underlying mechanisms of PTSD and the potential for targeted interventions.

## 6. Hypothalamic–Posterior Pituitary Axis and PTSD

### 6.1. Oxytocin and PTSD

Oxytocin (OXT) is a hormone the hypothalamus produces that is involved in various physiological and behavioral processes, including social bonding, trust, and stress regulation [196]. Oxytocin is synthesized as an inactive precursor protein from the oxytocin gene within the paraventricular and supraoptic nuclei of the hypothalamus. The active neuropeptide is then released from the posterior pituitary gland into the systemic circulation [197]. Oxytocin has multiple physiological functions, ranging from labor contractions and milk ejection [198] to promoting social communication between conspecifics [196] and possessing anti-inflammatory properties that make it a potent modulator of the immune system [199,200]. Oxytocin may exert an anxiolytic effect by modulating anxiety behavior and fear processes [201], leading to a decreased HPA axis response [202]. Increasing evidence suggests that oxytocin is involved in PTSD-associated behaviors, as it modulates cognitive impairment induced by stress, facilitates avoidance response extinction, and inhibits the action of CRH during stress response [203], all contributing to PTSD neurobiology. Converging evidence suggests the possible role of oxytocin in several aspects of PTSD pathophysiology, as exogenous oxytocin therapy has been shown to modulate fear responses and anxiety symptoms arising from aversive experiences [47]. 

Endogenous oxytocin levels in men and women show different expressions after trauma exposure, with lower levels of oxytocin observed in traumatized men [204]. In contrast, women generally demonstrate higher levels of endogenous oxytocin [205], which may help to reduce stress reactivity and motivate social contact-seeking behavior [206]. Supporting this finding, Nishi et al. reported that higher oxytocin levels in women exposed to trauma correlated with cooperativeness and seeking social support [207]. Decreased plasma oxytocin levels have been observed in individuals who developed PTSD, regardless of sex, suggesting a possible role for oxytocin in the disorder’s pathophysiology [208]. For example, a study exploring variations in salivary oxytocin reported lower oxytocin in Dutch male police officers with PTSD following a severe trauma compared to their healthy colleagues [204]. These findings suggest that decreased oxytocin levels may be an unspecific biomarker in chronic stress conditions [209].

Due to its anxiolytic and prosocial properties, oxytocin has been proposed as a promising pharmacological treatment for PTSD [210]. Research on the therapeutic potential of oxytocin suggests that intranasal administration of oxytocin may modulate the reactivity of brain areas associated with PTSD pathophysiology. Current intranasal oxytocin interventions have revealed that oxytocin administration can alter neural fear processing and thus might be a promising early preventive treatment for PTSD [204]. Interestingly, a meta-analysis of 66 magnetic resonance imaging (fMRI) studies found that amygdala activity decreased following intranasal administration of oxytocin [211]. The direct influence of oxytocin depends on targeting its receptors on CRH hypothalamic neurons, which affects subsequent HPA axis function [212]. These findings suggest that oxytocin may hold promise as a potential therapy for PTSD by altering neural activity in brain regions involved in fear and anxiety processing. 

A placebo-controlled trial investigated whether intranasal administration of oxytocin in the early stages of trauma would attenuate later clinical PTSD symptom severity. However, the findings indicated that oxytocin administration had no significant effect on the Clinician-Administered PTSD Scale (CAPS) compared to healthy controls during the post-trauma period [213]. Further analysis using fMRI data revealed that the bilateral connectivity between the right basolateral amygdala (BLA) and the dorsal anterior cingulate cortex (dACC) was enhanced in women with PTSD and that this connectivity was dampened following oxytocin administration [210]. Another study conducted by Flanagan et al. involved a randomized, placebo-controlled, double-blind design and demonstrated lower PTSD symptoms and depression following long-term exposure therapy with weekly oxytocin doses in affected patients [214]. Table 1 shows the current therapeutic strategies for PTSD patients correlated with oxytocin.

In general, individuals with PTSD may have lower levels of oxytocin than healthy individuals. Additionally, lower oxytocin levels have been associated with more severe PTSD symptoms, including anxiety, depression, and hyperarousal, suggesting that changes in oxytocin levels may contribute to developing and maintaining PTSD symptoms. Oxytocin also regulates stress response, learning and memory, emotion, reward, sleep, and wakefulness, which is often disrupted in individuals with PTSD. So, it may regulate various physiological and behavioral processes relevant to PTSD. For example, it affects social bonding and trust, and alterations in oxytocin levels may play a role in developing social isolation and avoidance behaviors observed in PTSD patients. However, although the exact mechanisms by which oxytocin affects the development of PTSD are not yet fully understood, research has suggested that alterations in oxytocin levels may contribute to the dysregulation of the stress response, the development of social isolation and avoidance behaviors, and the disruption of sleep in individuals with PTSD.

### 6.2. Vasopressin and PTSD

Vasopressin (AVP) is secreted by the hypothalamic dendrites, BnST, and amygdala [47,48,62], where it binds to vasopressin receptor 1A (V1A) in the septum, hippocampus, BnST, and other brain regions, as well as vasopressin receptor 1B (V1B), modulating social and stress-related memory, and likely emotionality [47,48]. As mentioned earlier, CRH and AVP in the hypothalamus work together to stimulate the anterior pituitary gland to release ACTH [63]. Alterations in central AVP signaling have been strongly suggested as contributing to various neuropsychiatric diseases, particularly those associated with anxiety, irregular fear, disturbed social behaviors, aggression, and PTSD [216]. 

Several studies have investigated changes in AVP signaling in patients with PTSD. AVP has long been proposed as a main trigger of the endocrine stress response [216], contributing to HPA axis activity as a co-stimulator that may be associated with PTSD [217]. Brain regions such as the PFC, hippocampus, and BLA have been implicated in PTSD [218]. The high density of V1A, a major vasopressin receptor in PTSD-related brain regions, supports the possible direct impact of AVP as a neurotransmitter in innervating brain areas involved in PTSD [219]. Studies have suggested that CA2 signaling is involved in social aggression via V1B receptors [220] and that central AVP signaling is related to behavioral alterations associated with PTSD-like symptoms via the V1A subtype [221]. Targeting AVP signaling by blocking the V1B receptors within the hypothalamus, especially the paraventricular nucleus (PVN), may be a promising therapeutic intervention for affective disorders [215].

## 7. Hypothalamus, Growth Hormone, and PTSD

Growth hormone (GH), also known as somatotropin, is a single-chain 22-kDa protein consisting of 191 amino acids. It is secreted from the anterior pituitary somatotropic cells and is involved in various physiological activities, including the stress response [222]. GH secretion displays a pulsatile pattern that is regulated by two hypothalamic hormones secreted into the hypophyseal portal system—somatostatin (SST) and GH-releasing hormone (GHRH) [223,224]. Somatostatin, also known as somatotropin release-inhibiting factor (SRIF), is produced by neurons located in the paraventricular (PVH) and periventricular (PV) nuclei. It inhibits the secretion of GH and GHRH produced by neurons located in the arcuate nucleus of the hypothalamus (ARC) and stimulates GH release. However, GH secretion is also influenced by other hormonal factors. For example, ghrelin, a GH-releasing peptide, is a potent endogenous GH-secretagogue (GHS) [204]. It can cross the blood–brain barrier (BBB) [225]. It can cross the blood–brain barrier (BBB) [226] and bind to the GHS receptor 1a (GHSR1a), also known as the ghrelin receptor. 

Ghrelin stimulates GH secretion by activating the GHS receptor in the hypothalamus and pituitary gland [227]. Stress exposure modulates ghrelin levels [226]. The GHS receptor, GHSR1a, is present in the BLA, which regulates fear. Ghrelin receptors appear to influence fear memory formation [228]. The source of ghrelin that modulates fear is unknown. Ghrelin is mainly synthesized by the stomach, with a smaller amount released by the small intestine [229]. However, much lower amounts have been found in several other tissues, such as the pancreas, lungs, and kidneys [230]. Additionally, a small number of neurons in the hypothalamus, cerebral cortex, and brainstem may produce ghrelin, acting as a paracrine hormone instead of being secreted into the blood [231]. Peripherally-derived ghrelin appears to play the dominant role in fear, although centrally-derived ghrelin remains a possible contributor [232]. Other situations that enhance GH release include fasting [212], hypoglycemia [233], hypoglycemia [234], exercise [235], puberty [236], and pregnancy [237]. GH has both central and peripheral effects, such as neurotropic effects [238], regulating metabolism through stimulating lipolysis in adipose tissue, food intake [239], glucose homeostasis [234], and influencing height, protein synthesis, and tissue growth [238].

The plasma half-life of GH is short, as evidenced by its pulsatile secretion. GH secretion induces the hepatic generation of insulin-like growth factor 1 (IGF-1), also known as somatomedin-C [240], which has greater stability due to its binding to IGF-1 binding proteins in circulation [241]. IGF-1 mediates several GH effects, and the GH-IGF-1 axis is critical in controlling somatic growth [238], achieved by regulating cellular division, regeneration, and proliferation in various tissues [242].

GH’s effects on the central nervous system (CNS) are mediated by the activation of brain GH receptors (GHRs). Numerous studies indicate the existence of GHRs in the brain, supporting the potential role of GH in CNS physiology [243,244]. The expression of GHR in the brain is essential for enabling neuroendocrine neurons to detect GH levels and regulate the secretion of pituitary GH through negative feedback [15]. Among all brain structures expressing GHR, the hypothalamus has the highest density of GH-responsive neurons. Thus, a significant proportion of GHR is expressed in the arcuate, paraventricular, and periventricular hypothalamic nuclei [245,246]. GHR belongs to the type I cytokine receptor family and primarily relies on the Janus kinase 2/signal transducer and activator of transcription (JAK2/STAT) pathway to exert its intracellular effects [242]. Central GHR signaling regulates various physiological functions, including metabolism, hormone secretion, stress response, spatial memory, and fear memory, by influencing specific brain regions such as the hypothalamic arcuate and ventromedial nuclei, hypothalamic paraventricular nucleus, hippocampus, and amygdala, respectively [238]. 

The amygdala contains more GH than any other brain region besides the pituitary gland [247]. In the amygdala, the ghrelin-GH axis regulates fear memory formation, potentially contributing to excessive fear memory typical of PTSD (Figure 4) [232,248]. Amygdala hyperactivity is observed in animals with chronic stress and patients suffering from trauma-related disorders [232,249], and it is likely the site of fear-increasing effects of repeated stimulation of ghrelin receptors. Increased activity in ghrelin receptors is sufficient and required for stress-induced fear and is independent of HPA axis activity. Repeatedly activating ghrelin receptors in stress-free animals increases fear memory without elevating HPA axis stress hormones. In contrast, systemic blocking of the ghrelin receptor during chronic stress inhibits the increase in stress-related fear, even at high levels of adrenal stress hormones. GH, a downstream mediator of ghrelin in the amygdala, is elevated in the BLA following chronic stress and increased fear memory strength [232]. Virus-mediated overexpression of GH in the amygdala enhances fear, an effect also observed in ghrelin-treated animals. Elevated levels of GH increase the number of amygdala neurons activated during fear memory formation and enhance the immediate early gene (IEG) c-Fos expression in the amygdala following fear learning. Additionally, GH increases dendritic spine density in pyramidal neurons of BLA. These findings provide a specific mechanism by which prolonged stress, through upregulation of GH in the BLA, may promote the development of PTSD [248].

A human study also demonstrated that individuals with PTSD had significantly higher levels of serum GH than normal controls [222]. The elevated levels of GH in PTSD could be due to the interaction between neurotransmitters and neuroendocrine systems. However, another study showed reduced GH levels and increased awakening during sleep in veterans with PTSD [250]. The decline in GH secretion may be linked to sleep fragmentation in these patients. It has been suggested that the quality of sleep and reduced nocturnal secretion of GH may impair cognitive functioning. The decline in GH secretion may be linked to sleep fragmentation in these patients. It has been suggested that the quality of sleep and reduced nocturnal secretion of GH may impair cognitive functioning [250]. One possible explanation for the difference in GH secretion in individuals with PTSD is that the hormonal changes resulting from traumatic events may depend on factors such as the severity and duration of exposure to stress, the amount of time elapsed since the traumatic event, and the individual’s coping strategies.

In conclusion, growth hormone (GH) is critical in various physiological activities, including the stress response. GH secretion is regulated by two hypothalamic hormones, somatostatin and GH-releasing hormone, and is influenced by other hormonal factors, such as ghrelin. The amygdala, which contains more GH than any other brain region besides the pituitary gland, plays a crucial role in fear memory formation, potentially contributing to excessive fear memory typical of PTSD. Elevated levels of GH in the amygdala following chronic stress and increased fear memory strength provide a specific mechanism by which prolonged stress may promote the development of PTSD. However, the hormonal changes resulting from traumatic events may depend on factors such as the severity and duration of exposure to stress, the amount of time elapsed since the traumatic event, and the individual’s coping strategies. Further research is needed to fully understand the complex interactions between the hypothalamus, GH, and PTSD.

## 8. Hypothalamus, Prolactin, and PTSD

Prolactin is a hormone responsible for stimulating milk production in females after childbirth. The biological actions of prolactin are not limited to reproduction because it has been shown to control various behaviors and even play a role in homeostasis. The stimuli that elicit prolactin release are not limited to nursing alone; instead, they may include exposure to light, sound, olfaction, and stress, all of which may have a stimulatory effect. Although it is well known that dopamine of hypothalamic origin provides inhibitory control over the secretion of prolactin, other factors within the brain, pituitary gland, and peripheral organs have been shown to inhibit or stimulate prolactin secretion as well [251,252,253,254,255]. Several hormones control the secretion of prolactin from the pituitary gland, such as dopamine (prolactin-inhibiting factor, PIF), serotonin, estradiol, progesterone, TRH, and vasoactive intestinal peptide (VIP) [256,257]. The hypothalamus releases dopamine which inhibits prolactin. However, the hypothalamus is involved in the acute stimulatory control of prolactin secretion by removing the inhibition (disinhibition) and superimposing brief stimulatory input. Prolactin secretion is also subject to the influence of multiple factors released either by the lactotrophs (autocrine regulation) or other cells within the pituitary gland (paracrine regulation) [252].

Prolactin is a hormone primarily associated with lactation, but it is also involved in various physiological processes, including stress regulation. It modulates emotionality and HPA axis reactivity [258]. Prolactin may counteract the immunosuppressive impact of inflammatory mediators during stress [259,260] and exert anxiolytic and anti-stress effects [258]. In addition to T and B cells, prolactin stimulates natural killer cells, neutrophils, macrophages, CD34+ hematopoietic cells, and antigen-presenting dendritic cells [260]. Conversely, prolactin secretion is affected by stress. Prolactin may play a role in stress-induced dysfunction and intestinal inflammation [261]. Therefore, the release of prolactin as a response to stress is of considerable clinical importance. Several stress models have been used to characterize such effects on prolactin secretion, including ether stress [262,263,264], restraint stress [265,266,267], thermal stress [268], hemorrhage [269], social conflict [270], migraine [271], and even academic stress in humans [272,273]. 

Stress and prolactin have been noticed for several decades. López-Calderón et al. reported in 1984 that stress decreases prolactin levels [274]. However, subsequent results were contradictory. Tomei et al. demonstrated that prolactin levels were higher in male and female police officers exposed to various chemical and urban stressors [275]. Similarly, Lennartsson et al. showed elevated serum prolactin levels following acute psychosocial stress. Although they found no differences between men and women, they have alleged that the women may have higher prolactin levels in response to stressors attributed to higher estradiol levels [276]. Even an exciting but stressful experience such as parachuting can increase prolactin levels [277]. Low prolactin levels may be associated with an increased risk of suicidal behavior, reported in males and females [278,279].

Like stress, the correlation between PTSD and prolactin has yielded inconclusive findings. Some studies report increased prolactin levels, but others demonstrate no changed or decreased values. Studies have shown that plasma prolactin levels in patients with PTSD can either increase [280], remain stable [222], or decrease [142,281] depending on various factors. For example, while Vidović et al. reported that prolactin levels remain significantly elevated in PTSD patients [280], Dinan et al. found no significant differences in prolactin levels between individuals with PTSD and healthy controls following buspirone administration [282]. Similarly, Schweitzer et al. conducted a study on veterans diagnosed with PTSD and reported no differences in baseline and peak prolactin compared to control groups following a d-fenfluramine challenge test [283]. Song et al. found no alteration in prolactin values of earthquake survivors with PTSD [222]. Again, Bauer et al. reported no differences between prolactin levels in refugees and healthy subjects [111]. However, Jergović et al. found an association of chronic PTSD with lower prolactin levels [281]. In addition, reduced serum prolactin is reported following an animal model of maternal PTSD [284]. Moreover, some studies alleged that dysregulation of the HPA and HPT axes might affect various hormones, leading to low prolactin levels [285].

In summary, the correlation between prolactin and PTSD is not clear. Some studies have reported lower prolactin levels in PTSD patients, while others demonstrated no differences or even higher prolactin values. Further research is needed to understand the relationship between prolactin and PTSD fully.

## 9. Conclusions and Future Directions 

PTSD is a significant psychiatric disorder that affects a considerable number of people worldwide. It typically arises following traumatic experiences, and some individuals may be more prone to it than others. The current treatments available for PTSD patients include cognitive behavioral therapy (CBT) as well as medication. Selective serotonin reuptake inhibitors, commonly used as antidepressants, are the primary drug choice for treating PTSD. However, these treatments are often ineffective and may cause side effects. Therefore, understanding the mechanisms involved in the development of this disorder is crucial for effective treatment.

Research suggests that in PTSD, several hypothalamic–pituitary–peripheral gland axes, including the HPA, HPG, and HPT axes, as well as posterior pituitary hormones, such as oxytocin and vasopressin, prolactin, and growth hormone, exhibit varying degrees of abnormalities. Each of these abnormalities may play an essential role in the development of PTSD and the emergence of its symptoms, as illustrated in Figure 5. Therefore, investigating these mechanisms further may provide insights into potential new treatments for PTSD.

Therefore, one of the top priorities for future clinical and preclinical research in PTSD is to focus critically on the role of various hormones in the hypothalamic–pituitary axis and the brain regions affected by these hormones in the development of PTSD. This approach will help us understand the role of these hormones in the pathophysiology of PTSD, and it may open up new avenues for designing novel drugs and treatment methods to manage this condition. Additionally, it is crucial to note that our attention and focus should not be solely on the HPA axis when treating PTSD patients. As stated in the article, other neuroendocrine disorders also significantly contribute to the development of this condition, and clinicians should consider this while evaluating and treating patients.

In preclinical research, it is important to develop more appropriate animal models with high predictive validity, face validity, and construct validity to explore the contribution of different neuroendocrine components of the hypothalamic–pituitary axis to the development of PTSD. These models could provide a more detailed understanding of the disease and aid in the development of effective treatments.

Finally, exposure-based protocols, including prolonged exposure (PE) therapy, are widely considered the gold standard in PTSD treatment, as they have been shown to effectively reduce PTSD symptoms. While CBT approaches can also be effective, they may not consistently produce successful outcomes across all subpopulations. Ongoing research efforts aim to enhance our understanding of PTSD, its treatment, and the factors that influence treatment outcomes in different subpopulations, including combat and non-combat veterans. By identifying the most effective treatment approaches for PTSD, we can improve the quality of life for those affected by this debilitating condition.

## Figures and Tables

**Figure 1 brainsci-13-01010-f001:**
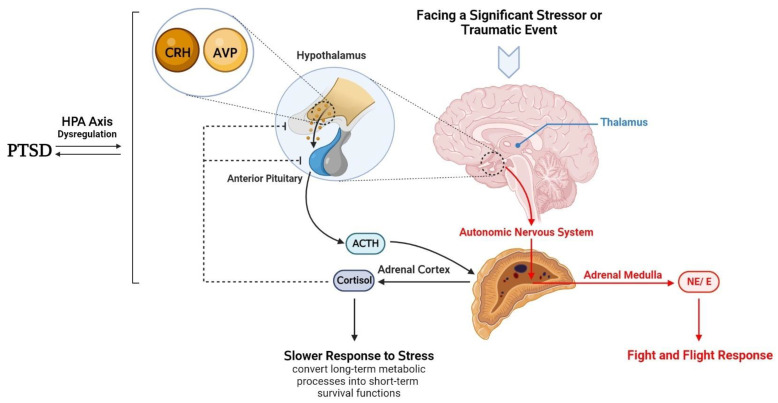
This figure depicts the hypothalamus’s function following exposure to significant stress or traumatic events. The hypothalamus activates two distinct pathways in response to stress: the ANS (specifically, the sympathetic nervous system) and the HPA axis. The sympathetic nervous system acts faster by releasing noradrenaline and adrenaline from the adrenal medulla, while the HPA axis pathway acts slower by releasing glucocorticoids from the adrenal cortex. CRH, primarily released from the hypothalamus, initiates the HPA axis. Along with AVP, CRH affects the anterior pituitary to release ACTH, which drives the adrenal cortex to produce glucocorticoid hormones such as cortisol in humans and corticosterone in rats, mice, birds, and most reptiles. Dysregulation of the HPA axis is associated with several mental disorders, including PTSD, depression, and schizophrenia. Understanding the complex interplay between the ANS and the HPA axis can provide insight into the physiological response to stress and the development of mental disorders. ANS: the autonomic nervous system; CRH: corticotropin-releasing hormone; HPA: hypothalamic–pituitary–adrenal; AVP: arginine vasopressin; ACTH: adrenocorticotropic hormone; PTSD: post-traumatic stress disorder; NE: norepinephrine, E: epinephrine.

**Figure 2 brainsci-13-01010-f002:**
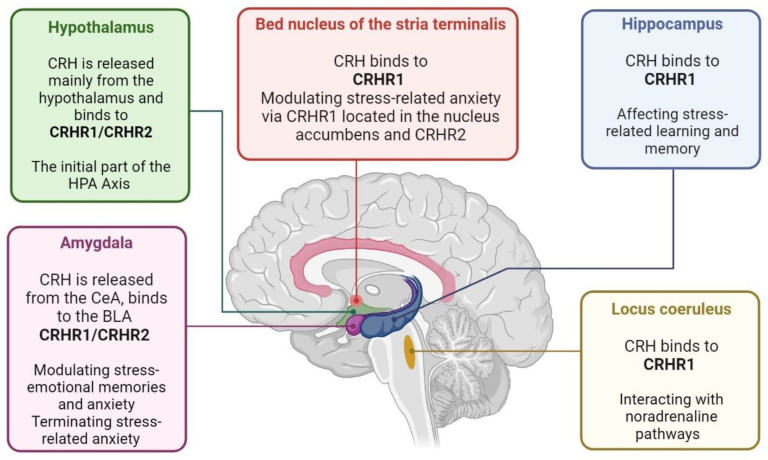
This figure depicts the brain regions that secrete CRH. While the hypothalamus is the primary source of CRH, other brain areas also secrete this hormone. CRH binds to its BLA, CRHR1, and CRHR2 receptors in the central amygdala, modulating stress-emotional memories and anxiety. Additionally, CRH is involved in terminating stress-related anxiety. In the locus coeruleus, CRH binds to CRHR1 and interacts with the noradrenaline pathway. In the hippocampus, CRH released by interneurons binds to pyramidal cells CRHR1, affecting stress-related learning and memory. The BnST releases CRH, which modulates stress-related anxiety via CRHR1 located in the nucleus accumbens and CRHR2. Understanding the distribution of CRH and its receptors throughout the brain provides insight into the complex role of this hormone in stress-related behaviors. CRH is a critical component of the HPA axis and plays a crucial role in the body’s response to stress. CRH: corticotropin-releasing hormone, CRHR: corticotropin-releasing hormone receptor. HPA: hypo-thalamic–pituitary–adrenal; CeA: central amygdala; BLA: basolateral amygdala; BnST: bed nucleus of the stria terminalis.

**Figure 3 brainsci-13-01010-f003:**
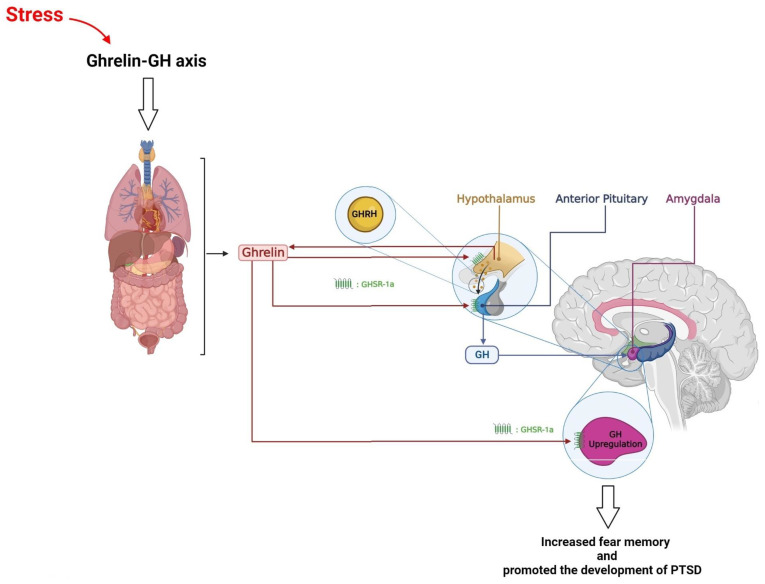
This figure illustrates the potential interaction between the ghrelin-GH axis and PTSD. Chronic stress increases ghrelin levels and activates the amygdala’s ghrelin receptors (GHSR1a). The enhanced activity of ghrelin receptors potentiates stress-induced fear memory, independent of the HPA axis activity. The source of ghrelin that modulates fear is not yet clear, as ghrelin is synthesized mainly in the stomach and released to a lesser extent by the small intestine, pancreas, lungs, kidney, hypothalamus, cerebral cortex, and brainstem. While peripherally-derived ghrelin appears to play a dominant role in fear memory, centrally-derived ghrelin may also contribute to this process. Ghrelin stimulates GH secretion by activating GHSR in the hypothalamus and pituitary gland. GH is a major downstream effector of ghrelin receptor activation and is increased in the amygdala following chronic stress. The joint action of ghrelin and GH in the amygdala leads to a significant increase in fear and may promote the development of PTSD. Understanding the complex interaction between the ghrelin-GH axis and PTSD may provide insight into the underlying mechanisms of this disorder and may lead to the development of more effective treatments. GH: growth hormone, GHSR: growth hormone secretagogue receptors, GHRH: growth hormone-releasing hormone, GHSR1a: growth hormone secretagogue receptor 1a, PTSD: post-traumatic stress disorder.

**Figure 4 brainsci-13-01010-f004:**
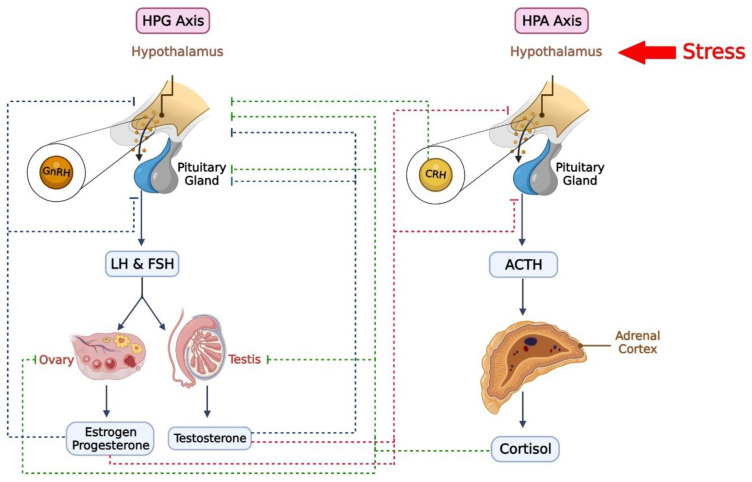
This figure illustrates the complex interactions between the HPA and HPG axes. In the HPA axis, CRH neurons in the hypothalamus’s PVN secrete CRH, stimulating corticotroph cells in the anterior pituitary to produce ACTH. ACTH affects the adrenal gland to produce cortisol in humans and corticosterone in rodents (right side of the picture). These hormones, via a negative feedback mechanism, prevent corticotrophs and PVN from secreting ACTH and CRH, respectively (not shown). In the HPG axis, GnRH neurons in the POA of the hypothalamus secrete GnRH, which stimulates gonadotroph cells in the anterior pituitary to produce LH and FSH. These hormones affect the gonadal glands, including the testes and ovaries, to produce gonadal steroid hormones such as testosterone, estrogen, and progesterone. Through a negative feedback mechanism, these hormones prevent the POA from secreting GnRH and gonadotrophs from secreting FSH and LH (left side of the picture). In stress conditions, the HPA axis exerts inhibitory effects on the HPG axis. Cortisol inhibits GnRH neurons in the hypothalamus, LH and FSH secretions of the pituitary gland, and the secretion of gonadal steroid hormones. In this inhibitory effect of the HPA axis, CRH also prevents GnRH secretion. The HPG axis can also modulate the HPA axis. Gonadal steroid hormones, such as testosterone, estrogen, and progesterone, inhibit CRH secretion by affecting the PVN of the hypothalamus and ACTH secretion by affecting the corticotrophs of the pituitary gland. Understanding the complex interactions between the HPA and HPG axes is crucial for understanding the physiological response to stress and regulating reproductive function. Blue lines show excitatory pathways, green dashed lines show the inhibitory effects of the HPA axis on the HPG axis, and red dashed lines show the inhibitory effects of the HPG axis on the HPA axis. HPA axis: hypothalamic–pituitary–adrenal axis; HPG axis: hypothalamic–pituitary–gonadal axis; CRH: corticotropin-releasing hormone; PVN: paraventricular nucleus; ACTH: adrenocorticotropic hormone; GnRH: gonadotropin-releasing hormone; POA: preoptic area; LH: luteinizing hormone; FSH: follicle-stimulating hormone.

**Figure 5 brainsci-13-01010-f005:**
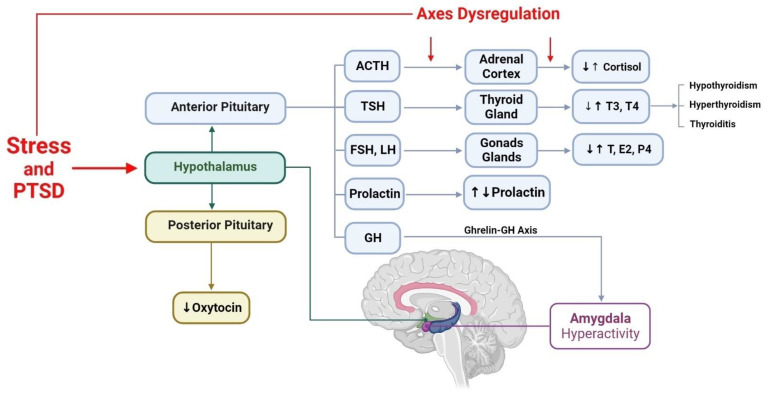
This figure provides an overview of the complex interplay between various neuroendocrine functions of the hypothalamus and stress-related disorders, focusing on PTSD. The figure highlights the involvement of possible dysregulated pathways and key hormones such as ACTH, TSH, FSH, LH, GH, testosterone, estradiol, and progesterone, which are critical for understanding the underlying mechanisms of PTSD and other stress/trauma-related disorders. PTSD: post-traumatic stress disorder; ACTH: adrenocorticotropic hormone; TSH: thyroid-stimulating hormone; FSH: follicle-stimulating hormone; LH: luteinizing hormone; GH: growth hormone; T: testosterone; E2: estradiol; P4: progesterone.

**Table 1 brainsci-13-01010-t001:** The current therapeutic strategies for PTSD patients correlated with OXT. OXT: oxytocin; BLA: basolateral amygdala; dACC: dorsal anterior cingulate cortex, CAPS: Clinician-Administered PTSD Scale; vmPFC: ventromedial prefrontal cortex; CeM: central nucleus of the medial amygdala.

Type of Study	Method	Mechanism	Findings	Ref.
A double-blind, placebo-controlled clinical trial	Intranasal oxytocin administration	Increased amygdala reactivity to fearful faces attenuated ventrolateral prefrontal cortex functional connectivity.	Repeated administration early post-trauma reduced later PTSD symptom development.	[204]
A meta-analysis study	Intranasal oxytocin administration	OXT binds to its receptors on CRH neurons.	Amygdala activity decreased in psychiatric populations and healthy ones.	[211]
A randomized controlled trial	Intranasal OXT (40 IU/dose, five puffs of 4 IU per nostril)	The exact mechanisms of the OXT effect in patients with high acute PTSD symptoms cannot be interpreted based on clinical data in this study.	Patients with high baseline CAPS scores had significantly lower CAPS scores after receiving oxytocin.	[213]
A randomized, placebo-controlled pilot trial	Intranasal OXT (40 IU)	Powered studies need to evaluate potential mechanisms.	Patients demonstrated lower PTSD and depression symptoms during long-term exposure without statistical significance.	[214]
A randomized controlled trial	Intranasal OXT	The control of the vmPFC over the CeM increased in males or via decreased dACC and BLA salience processing in females.	Anxiety and nervousness were decreased in PTSD patients.	[210]
An original study	AVP	Blocking the V1bR within the PVN.	Therapeutic intervention for PTSD.	[215]

## Data Availability

Not applicable.

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
