# Peer review of "Hypothalamus and Post-Traumatic Stress Disorder: A Review"

_brainsci, 2023, doi:10.3390/brainsci13071010_

Round 1

Reviewer 1 Report

Comments and Suggestions for Authors

I thank the authors for their interesting, thorough, and timely manuscript. I am have confined my comments to my domain of expertise. I urge the authors to update their introduction section to reflect the heterogeneity in outcomes for treatment and non-treatment seeking samples with PTSD. The Breslau article is over 20 years old – more recent research shows variability in outcomes based on intervention approach and subpopulation (e.g. combat and non-combat veterans, adult sexual assault survivors, those with childhood trauma, etc). There appears to be a word missing in this sentence “PTSD is the fourth psychiatric disorder…” I encourage the authors to more accurately reflect the state of the treatment literature in their conclusion. The gold standard therapies for PTSD are exposure-based protocols ((e.g. prolonged exposure) which are conceptually distinct from pure CBT approaches. PE and other exposure protocols are highly effective, though less so in veterans (see M. Steenkamp’s work on this topic).

Comments on the Quality of English Language

A copy edit is warranted but overall the English is of high quality

Reviewer 2 Report

Comments and Suggestions for Authors

In this review, Raise-Abdullahi and colleagues discuss the role of the hypothalamus in the pathophysiological mechanisms subserving Post-traumatic Stress Disorder.  The manuscript is interesting. However, some points should be addressed .

-       In the abstract: “ Stress is a highly subjective issue caused by any disturbance in people's daily activities, sometimes even terrible experiences such as war, rape, natural disasters, burns, car accidents, etc.”. This statement is questionable and unclear. The Authors should know that stress and trauma are different concepts (PMID: 34247187).

-       The manuscript is too long. There are many general concepts. The Authors must shorten the manuscript avoiding general concepts and repetitions that appear as copy and paste of previous statements.

-       The discussion of the link between circulating cortisol and PTSD is important. The Authors must to discuss better this part. I suggest to add more findings (although contrasting) from translational studies that have investigated this aspect (PMID: 28778658; PMID: 33392367; PMID: 29035710 and more).

Comments on the Quality of English Language

Moderate editing of English language is required. 

Round 2

Reviewer 2 Report

Comments and Suggestions for Authors

The Authors addressed all the points I raised.